# ATAC-seq footprinting unravels kinetics of transcription factor binding during zygotic genome activation

Mette Bentsen [1], Philipp Goymann [1], Hendrik Schultheis[1], Kathrin Klee [1], Anastasiia Petrova[1], René Wiegandt [1], Annika Fust[1], Jens Preussner [1,2], Carsten Kuenne [1], Thomas Braun [2,3], Johnny Kim [2,3] & Mario Looso [1,2 ✉]

While footprinting analysis of ATAC-seq data can theoretically enable investigation of transcription factor (TF) binding, the lack of a computational tool able to conduct different levels of footprinting analysis has so-far hindered the widespread application of this method. Here we present TOBIAS, a comprehensive, accurate, and fast footprinting framework enabling genome-wide investigation of TF binding dynamics for hundreds of TFs simultaneously. We validate TOBIAS using paired ATAC-seq and ChIP-seq data, and find that TOBIAS outperforms existing methods for bias correction and footprinting. As a proof-of-concept, we illustrate how TOBIAS can unveil complex TF dynamics during zygotic genome activation in both humans and mice, and propose how zygotic Dux activates cascades of TFs, binds to repeat elements and induces expression of novel genetic elements.

---

[1] Bioinformatics Core Unit (BCU), Max Planck Institute for Heart and Lung Research, 61231 Bad Nauheim, Germany. [2] German Centre for Cardiovascular Research (DZHK), Partner Site Rhine-Main, 60596 Frankfurt am Main, Germany. [3] Department of Cardiac Development and Remodeling, Max Planck Institute for Heart and Lung Research, 61231 Bad Nauheim, Germany. ✉email: mario.looso@mpi-bn.mpg.de

Epigenetic mechanisms governing chromatin organization and transcription factor (TF) binding are critical components of transcriptional regulation and cellular transitions. In recent years, rapid improvements of pioneering sequencing methods such as ATAC-seq (Assay of Transposase Accessible Chromatin)[1], have allowed for systematic, global scale investigation of epigenetic mechanisms controlling gene expression. While ATAC-seq can uncover accessible regions where TFs might bind, reliable identification of specific TF binding sites (TFBS) still relies on chromatin immunoprecipitation methods such as ChIP-seq. However, ChIP-seq methods require high input cell numbers, are limited to one TF per assay, and are further restricted to TFs for which antibodies are readily available. Therefore, it remains costly, or even impossible, to study the binding of multiple TFs in parallel.

Current limits to the investigation of TF binding become particularly apparent when investigating processes involving a very limited number of cells, such as preimplantation development (PD) and zygotic genome activation (ZGA) of early zygotes. Integration of multiple omics-based profiling methods have revealed a set of key TFs that are expressed at the onset of and during ZGA including Dux[2], Zscan4[3], and other homeobox-containing TFs[4]. However, due to the limitations of ChIP-seq, the exact genetic elements bound and regulated by different TFs during PD remain to be fully discovered. Consequently, the global network of TF binding dynamics throughout PD remains mostly obscure.

A computational method known as digital genomic footprinting (DGF)[5] has emerged as an alternative means, which can overcome some of the limitations of ChIP-based methods. DGF is a computational analysis of chromatin accessibility assays such as ATAC-seq, which employs DNA effector enzymes that only cut accessible DNA regions. Similarly to nucleosomes, bound TFs hinder cleavage of DNA, resulting in defined regions of decreased signal strength within larger regions of high signal—known as footprints[6] (Fig. 1a).

Surprisingly, although this concept shows considerable potential to survey genome-wide binding of multiple TFs in parallel from a single experiment, DGF analysis is rarely applied when investigating TF binding mechanisms. The skepticism towards DGF has been driven by the discovery that enzymes used in chromatin accessibility assays (e.g., DNase-I) are biased towards certain sequence compositions, an effect which has been well characterized for DNase-seq[7,8]. The influence of Tn5 transposase bias in the context of ATAC-seq footprinting has, however, only been described very recently[9,10] and still represents an uncertainty during discovery of true footprints. Besides the identification of footprints, comparing footprints across biological conditions remains challenging as well. While there have been efforts to estimate differential TF binding on a genome-wide scale[11,12], investigation of epigenetic processes often requires more in-depth information on the individual differentially bound TFBS and genes targeted by these TFs. Furthermore, many footprinting methods suffer from performance issues due to missing support for multiprocessing, inflexible software architecture, and the use of non-standard file-formats. These obstacles complicate the assembly of different tools for advanced analysis workflows. Consequently, despite its compelling potential, these issues have rendered footprinting on ATAC-seq cumbersome to apply to biological questions. Essentially, a comprehensive framework enabling large-scale ATAC-seq footprinting is missing.

Here, we describe TOBIAS (Transcription factor Occupancy prediction By Investigation of ATAC-seq Signal), a comprehensive computational framework that we created for footprinting analysis (Fig. 1b–f). TOBIAS is a collection of command-line tools utilizing a minimal input of ATAC-seq reads, TF motifs and genome information (Fig. 1b) to perform all levels of footprinting analysis including bias correction (Fig. 1c), footprinting (Fig. 1d), and comparison between conditions (Fig. 1e). Furthermore, TOBIAS includes a variety of auxiliary tools such as TF network inference and visualization of footprints, which allow for various downstream analysis (Fig. 1f, Supplementary Fig. 1). In this investigation, we apply TOBIAS to ATAC-seq data from both human and mouse PD and show how visible TF footprints correlate with the timings of TF activity throughout development. We additionally focus on the TF Dux, an important TF during ZGA, and use TOBIAS to unravel its target genes and influence on the global transcriptional network throughput PD.

## Results

**Impact of bias correction on footprint visibility.** To validate the results of the TOBIAS method, we utilized 217 paired ChIP-seq/ATAC-seq datasets across four different cell types (GM12878, A549, HepG2, and K562). Here, the ChIP-seq peaks represent the true binding sites for each TF, which we used for validating the accuracy of the binding sites predicted by footprinting (see Supplementary Methods part 3).

As it has been shown that the Tn5 transposase has a large effect on footprinting[10], the first step of the TOBIAS footprinting pipeline is Tn5 bias correction. The TOBIAS bias correction module (named ATACorrect) utilizes a dinucleotide weight matrix (DWM)[13] to estimate the background bias of the Tn5 transposase (Fig. 1c). This DWM is used to calculate an expected Tn5 signal for each genomic region, representing the influence of the Tn5 bias (Fig. 1c; expected cutsites). Subtracting these expected cuts from the uncorrected signals yields a corrected track, highlighting the effect of protein binding (details are available in Supplementary Methods part 1). In order to evaluate the performance of TOBIAS in comparison to existing bias correction tools, we utilized the paired ChIP-seq/ATAC-seq data mentioned above to visualize aggregated footprints across bound and unbound subsets of TFBS. We found TOBIAS to outperform other bias correction tools in uncovering footprints and thereby distinguishing between bound/unbound sites (Supplementary Fig. 2a, Supplementary Data 1). Next, we wanted to quantify the depths of the aggregated footprints and utilized a footprint depth (FPD) metric as described by Baek et al.[12] (Supplementary Fig. 2b). In line with the visual impression, TOBIAS has the most significant difference in FPD between bound and unbound subsets of TFBS (Supplementary Fig. 2c). Importantly, the FPD's of unbound sites are minimally affected by bias correction, indicating that bias correction only uncovers footprints for truly bound sites.

Of note, the TOBIAS 'ATACorrect' method relies on the calculation of the expected Tn5 cuts based on the influence of Tn5 bias. Interestingly, besides identifying cases where the footprint was hidden by Tn5 bias (Supplementary Fig. 2d; JDP2), the track of expected signal also identifies TFs for which the motif itself disfavors Tn5 integration, thereby creating a false-positive footprint in uncorrected signals (Supplementary Fig. 2d; FOXD3). We wanted to investigate this effect in more detail and found that there is a high correlation between the footprint depths of uncorrected and expected Tn5 signals across all TFs, which vanishes after TOBIAS correction (Supplementary Fig. 2e). This observation demonstrates that bias correction effectively uncovers TF footprints, which were otherwise superimposed by Tn5 bias. It has previously been suggested that only 20% of all TFs leave measurable footprints[12], and we were able to confirm this observation using the uncorrected footprint depths and the same metric (Supplementary Fig. 2f; uncorrected). However, in contrast, we observed a measurable footprint for 59% of the TFs

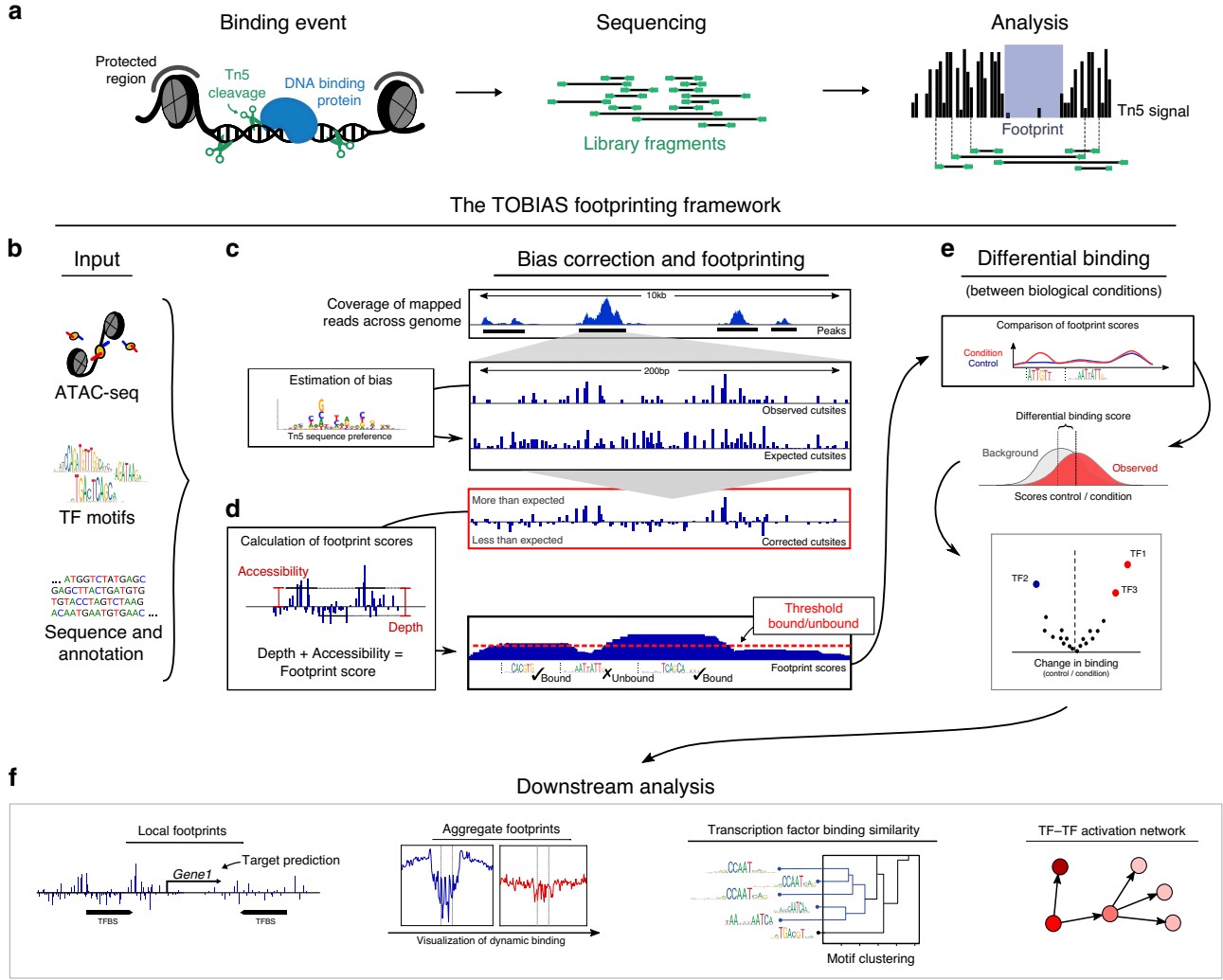

**Fig. 1 The TOBIAS digital genomic footprinting framework. a** The concept of footprinting using ATAC-seq. Tn5 transposase cleaves DNA and inserts sequencing adapters, but is unable to cut chromatin occupied by proteins such as nucleosomes (gray) and other DNA binding proteins e.g., transcription factors (blue). Sequencing libraries of DNA fragments are sequenced to yield reads (green). During analysis, each read is mapped to the genome and used to create a signal of single Tn5 insertion events (black bars), in which binding of protein is visible as depletion of the signal (defined as the footprint). **b** TOBIAS uses reads from ATAC-seq, transcription factor motifs and sequence annotation in standard formats as input. **c** Bias correction of Tn5 signal. In the first step, TOBIAS reads the observed Tn5 cutsites and estimates the underlying Tn5 sequence preference. TOBIAS then calculates the expected Tn5 cutsites per region, which represent the background probability of Tn5 insertion. Using the expected signal track, the Tn5 bias corrected cutsites are obtained (red box). **d** Footprinting to estimate transcription factor binding. The corrected cutsites enable calculation of footprint scores with a scoring function taking into account both the accessibility and depth of the local footprint (as depicted in the box labeled "Calculation of footprint scores"). This continuous footprint score is correlated with the presence of transcription factor binding sites in the genome, and a threshold is set to distinguish between bound and unbound sites. **e** Differential footprinting. If multiple conditions are investigated, the differential binding module summarizes individual site scores (upper black box) for each TF, and compares them between conditions (gray/red curve center) in order to define differentially bound TFs. Performed on all TFs under investigation, a volcano plot illustrates the global changes in transcription factor binding. **f** Additional analysis modules. After the main TOBIAS analysis, a variety of downstream analysis can be applied including visualization of local and aggregated footprints across conditions, comparison of binding specificity between individual transcription factors and TF network prediction.

when using the TOBIAS corrected signals (Supplementary Fig. 2f; corrected). As the fitted two-component model is a limited estimator to classify bound/unbound sites, we additionally calculated null distributions of randomized corrected footprints. By this approach, we similarly found the number of measurable footprints to be consistent at ~65% across all four cell types investigated (Supplementary Fig. 2g). This demonstrates that failure to correct for Tn5 bias can lead to false negative footprints, while bias correction uncovers the true amount of measurable footprints to be above 50%.

**Validation of TOBIAS footprinting**. For the task of protein binding prediction (i.e., footprinting), we collected four popular tools for ATAC-seq footprinting (HINT-ATAC, PIQ, Wellington, and msCentipede) and compared these to the individual TOBIAS framework features where applicable. While we found that some functionalities are overlapping between tools, we found a substantial set of features, such as differential footprinting for more than two conditions, to be exclusively covered by TOBIAS (Supplementary Table 1). Evaluating the results of each tool, we found that TOBIAS significantly outperformed the other de novo

tools HINT-ATAC, PIQ, and Wellington (Supplementary Fig. 3a) and performed equally well as msCentipede overall (Supplementary Fig. 3b). Notably, TOBIAS also showed robust performance across individual cell types (Supplementary Fig. 3c). Looking at individual TFs, TOBIAS outperforms msCentipede for factors such as CEBPB, which has a notable gain of footprints after Tn5 bias correction (Supplementary Fig. 3d), once again highlighting the advantage of taking Tn5 bias into account. Although msCentipede implements a motif centric learning approach, which can take TF specific binding patterns into account, it did not yield overall higher accuracy in comparison to TOBIAS. Additionally, the approach of building individual TF models took 300 times longer to compute than performing footprinting using TOBIAS (Supplementary Fig. 3e). Such learning approaches are therefore greatly limited in the number of TFs and conditions, which can realistically be analyzed.

Although we have shown that more than half of TFs create visible aggregated footprints, the footprints at individual loci are much more difficult to detect due to the sparsity of the ATAC-seq signal (as seen in Fig. 1f; Local footprints). In order to take this sparsity into account, we have designed the TOBIAS footprinting score as a combined score taking into account both depletion and accessibility (Supplementary Fig. 3f). In comparison, previous scoring methods such as the Footprint Occupancy Score (FOS)[14], calculate the difference in signal level between the background and the footprint (Supplementary Fig. 3g). To test the impact of this novel scoring approach, we compared the results of the TOBIAS (depletion + accessibility) score (as calculated from corrected cutsites) with the FOS score (pure depletion) (Supplementary Fig. 3h). While there is a limited improvement in the FOS footprinting score by using TOBIAS corrected cutsites, we found that there is a significant increase in predictive ability by using the TOBIAS score. This shows that, although bias correction is highly important for visualizing aggregated footprints, the influence of accessibility in the calculation of footprint scores is of considerable importance as well. Along this line, these findings illustrate the relationship between aggregated footprints and individual TFBS footprints. While the number of TFs with footprints in aggregated signals is above 50%, the proportion of individual TFBS supported by footprints might be considerably lower. Consequently, a score like FOS, which requires a footprint depletion for prediction, is inherently limited when predicting protein binding. In conclusion, we found that TOBIAS exceeded other tools in terms of uncovering footprints hidden by bias and correctly identifying bound TF binding sites. The improvement in accuracy is achieved by the alternative approaches for bias correction as well as by the novel footprinting score.

**TF binding dynamics in mammalian ZGA.** To demonstrate the potential of TOBIAS to predict differential TF binding across multiple conditions, in particular in the investigation of processes involving only few cells, we analyzed a series of ATAC-seq datasets derived from both human and murine preimplantation embryos at different developmental stages ranging from 2C, 4C, 8C to ICM in addition to embryonic stem cells of their respective species[15,16] (Fig. 2a). Altogether, TOBIAS was used to calculate footprint scores for a list of 590 and 464 individual TFs across the entire process of PD of human and mouse embryos, respectively. After clustering TFs into co-active groups within one or multiple developmental timepoints, we first asked whether the predicted timing of TF activation reflects known processes in human PD. Intriguingly, we found 10 defined clusters of specific binding patterns, the majority of which peaked between 4C and 8C, fully concordant with the transcriptional burst and termination of ZGA (Fig. 2b).

Two clusters of TFs (Cluster 1 + 2; $n = 83$) displayed highest activity at the 2–4C stage and strongly decreased thereafter, suggesting that factors within these clusters are likely involved in ZGA initiation. We set out to classify these TFs, and observed a high overlap with known maternally transferred transcripts[17] (LHX8, BACH1, EBF1, LHX2, EMX1, MIXL1, HIC2, FIGLA, SALL4, and ZNF449), explaining their activity before ZGA onset. Importantly, DUX4 and DUXA, which are amongst the earliest expressed TFs during ZGA[2,18], were also contained in these clusters. Additional TFs included HOXD1, which is known to be expressed in human unfertilized oocytes and preimplantation embryos[19] and ZBTB17, a TF mandatory to generate viable embryos[20]. Cluster 6 ($n = 67$) displayed a particularly prominent 8C specific signature, that harbored well-known TFs involved in lineage specification such as PITX1, PITX3, SOX8, MEF2A, MEF2D, OTX2, PAX5, and NKX3.2. Furthermore, overlapping TFs within Cluster 6 with RNA expression datasets ranging from the germinal vesicle to cleavage stage[2], 12 additional TFs (FOXJ3, HNF1A, ARID5A, RARB, HOXD8, TBP, ZFP28, ARID3B, ZNF136, IRF6, ARGFX, MYC, and ZSCAN4) were confirmed to be exclusively expressed within this time frame. Taken together, these data show that TOBIAS reliably uncovers massively parallel TF binding dynamics at specific timepoints during early embryonic development.

**TF binding correlates with visible footprints.** To confirm that TOBIAS-based footprinting scores are indeed associated with leaving bona fide footprints, we utilized the ability to visualize aggregated footprint plots as implemented within the framework. Indeed, bias corrected footprint scores were highly congruent with explicitly defined footprints (Fig. 2c) of prime ZGA regulators at developmental stages in which these have been shown to be active[3]. For example, footprints associated with DUX4, a master inducer of ZGA, were clearly visible from 2C–4C, decreased from 8C onwards and were completely lost in later stages, consistent with known expression levels[15] and ZGA onset in humans. Footprints for ZSCAN4, a primary DUX4 target[2], were exclusively visible at the 8C stage. Interestingly, GATA2 footprints were exhibited from 8C to ICM stages which is in line with its known function in regulating trophoblast differentiation[21]. As expected, CTCF creates footprints across all timepoints. Strikingly, we observed that these defined footprints were not detectable without TOBIAS-mediated Tn5 bias correction (Supplementary Fig. 4a). These data show that footprint scores can be reliably confirmed by footprint visualizations, which further allow to infer TF binding dynamics.

To test if the global footprinting scores of individual TFs correlate with the incidence and level of their RNA expression, we matched them to RNA expression datasets derived from individual timepoints throughout zygotic development, taking TF motif similarity into account. Indeed, we found that TOBIAS scores for the majority of TFs either correlated well with the timing of their expression profiles or displayed a slightly delayed activity after expression peaked (Supplementary Fig. 4b). This is important because it shows that in conjunction with expression data, TOBIAS can indicate the kinetics between TF expression (mRNA) and the actual binding activity of their translated proteins. The value of this added information becomes particularly apparent when analyzing activities of TFs that did not correlate with the timing of their RNA expression (Supplementary Fig. 4b; not correlated).

For example, within the non-correlated cluster 13 TFs were identified, which are of putative maternal origin[17] including SALL4. In mice, Sall4 protein is maternally contributed to the zygote, subsequently degraded at 2C and then re-expressed after

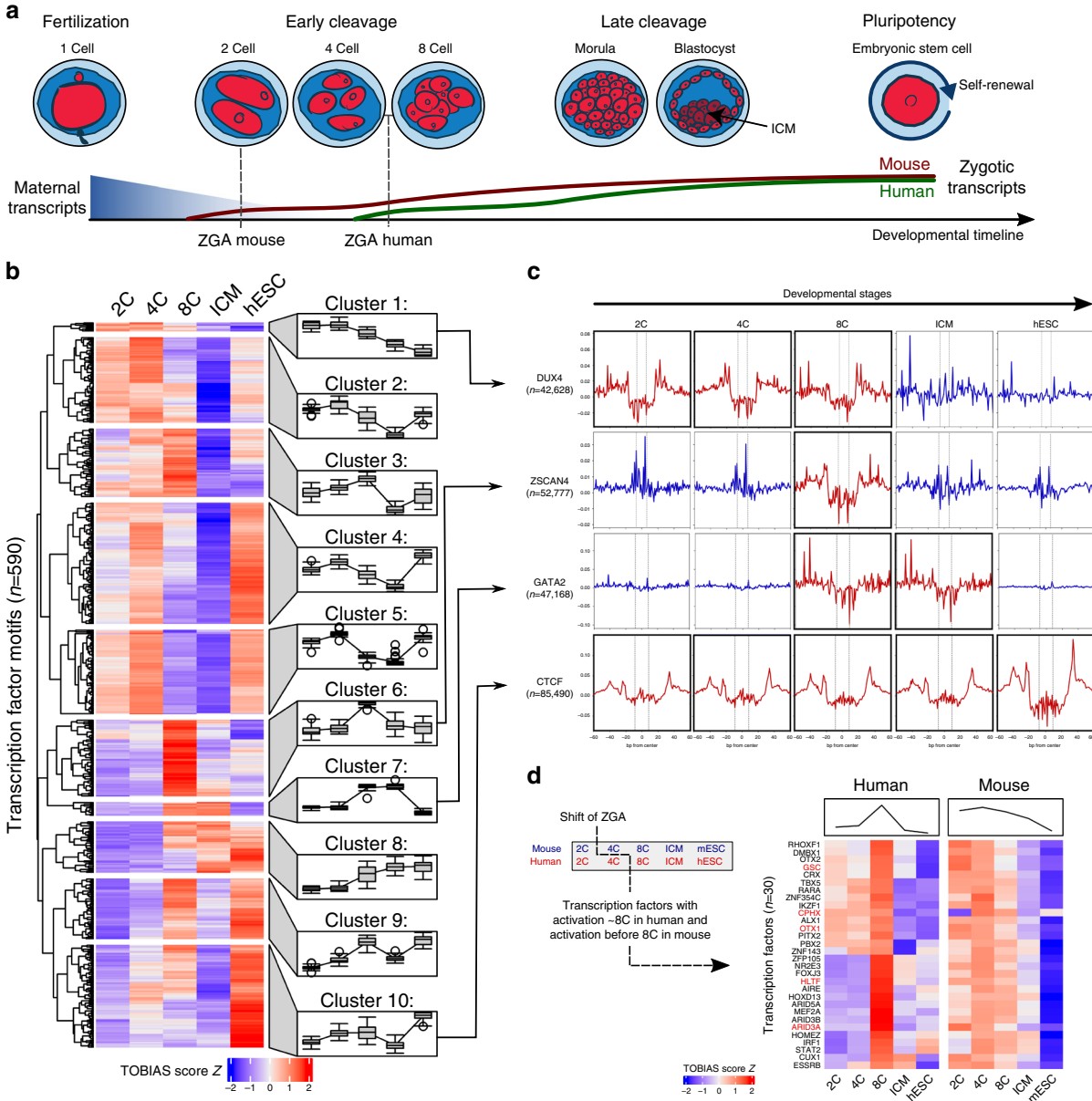

**Fig. 2 Global changes in transcription factor binding throughout embryonic development. a** Early embryonic development in human and mouse. While the fertilized egg undergoes a series of divisions, the maternal transcripts (blue curve) provided by the egg are depleted over time, and the zygotic genome is activated in waves (ZGA, red and green lines). ZGA initiates at the 2-cell stage in mouse and at the 4–8-cell stage in human. **b** Clustering of transcription factor activities throughout development. Each row represents one TF, each column a human developmental stage. TF activity scores from TOBIAS are Z-score transformed across rows. Blue color indicates low activity, red color indicates high activity. In order to visualize cluster trends, each cluster is associated with a mean trend line (left to right) and timepoint specific boxplots respectively. Source data are available in the Source Data file. **c** Bias corrected ATAC-seq footprints. For selected TFs with known roles in early development originating from four clusters (arrows from **b**), an aggregated footprinting plot matrix for all associated transcription factor binding sites is shown. Individual plots are centered around binding motifs (n = asterisk (*) relates to the number of binding sites). Rows indicate TFs DUX4, ZSCAN4, GATA2, and CTCF; columns illustrate developmental stages from left to right. Active binding of the individual TFs is visible as depletion in the signal around the binding site (highlighted in red). See Supplementary Figure 4a for corresponding uncorrected footprints. **d** TF activity onset in human and mouse. Heatmaps show activity of known ZGA-related TFs for human (left) and mouse (right) across matched timepoints 2C/8C/ICM/hESC (mESC). Transcription factors with known roles in ZGA are highlighted in bold red.

zygotic transcription has initiated[22]. Consistent with this, *SALL4* expression increases dramatically from 8C onwards (see Source Data file). In contrast to the expression values, TOBIAS predicted SALL4 to have the highest activity in 2C, with decreasing activity in 8C, which is in line with the presence of maternal SALL4 in the zygote. Comparing this change to all TF changes between 2C and 8C (log2 fold-changes estimated from TOBIAS activity scores),

we find that SALL4 is at the 7th percentile of all changes ranked from decreasing to increasing, which is consistent with the degradation of the protein after the 2C stage. These data show that TOBIAS can provide significant insight into TF activities, in particular for those where determining their expression patterns alone does not suffice to explain when they exert their biological function.

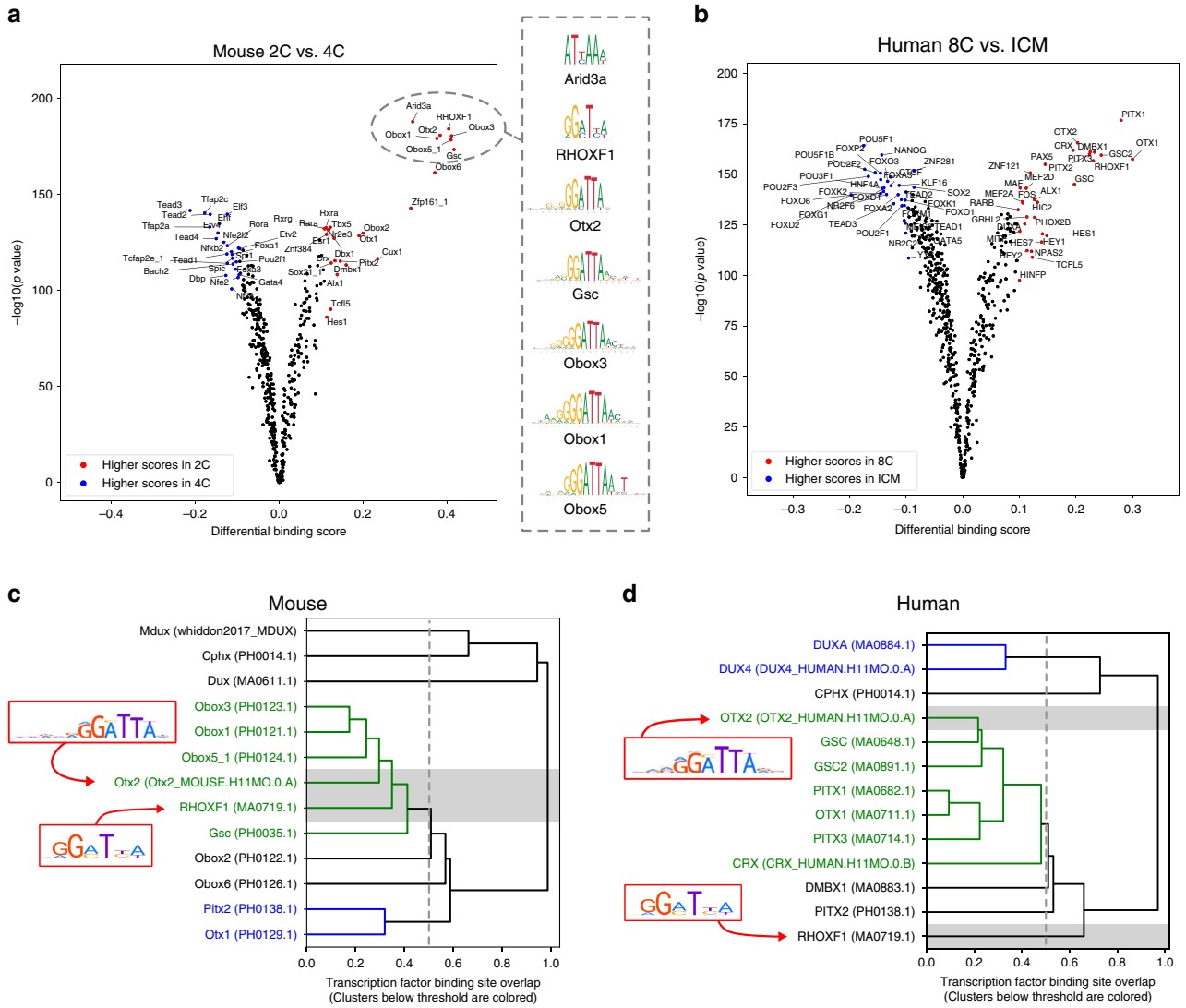

**Fig. 3 Specification of ZGA functions between mouse and human. a, b** Pairwise comparison of TF activity between developmental stages. The volcano plots show the differential binding activity against the −log10(*p* value) (both provided by TOBIAS) of all investigated TF motifs; each dot represents one motif. For **a** 2C stage specific TFs are labeled in red, 4C specific factors in blue. From the 2C specific TFs, seven prominent examples are chosen and illustrated by their motif. For **b** 8C stage specific TFs are labeled in red, ICM specific factors in blue. **c, d** Clustering of TF motifs based on binding-site overlap. Excerpt of the global TF clustering based on TF binding location, illustrating individual TFs as rows. The trees indicate genomic positional overlap of individual TFBS. A tree depth of 0.2 represents an overlap of 80% of the motifs. Each TF is indicated by name and unique ID in brackets. Clusters of TFs with more than 50% overlap (below 0.5 tree distance) are colored in green/blue. The position of TF motifs RHOXF1 and Otx2/OTX2 are highlighted. **c** shows overlap of motifs included in the mouse analysis. **d** shows clustering of human motifs. Complete TF trees are provided in Supplementary Notes 1 and 2.

**Comparison of TF binding between human and mouse ZGA.** The timing of ZGA varies between mice (2C) and humans (4C–8C) (reviewed in ref. [23]). By integrating the TOBIAS scores from human and mouse (Supplementary Fig. 4c), and instrumentalizing the capability of TOBIAS to generate differential TF binding plots for all timepoints automatically, we investigated similarities and differences of PD between these species. Firstly, reflecting the shift of ZGA onset, we identified 30 TFs, which appeared to be ZGA specific in both human and mouse (Fig. 2d), including OTX1, GSC, CPHX, and HLTF, which already have described functions within ZGA[4,24]. Moreover, this list also includes ARID3A, which has been shown to play a role in cell fate decisions in creating trophectoderm[25].

Next, we wanted to investigate specific differentially bound TFs, not only across the whole timeline, but also between individual conditions. We therefore utilized the differential TF

binding plots created by TOBIAS, and chose to focus on the cellular transition initiated at and following ZGA, which corresponds to the transition between 2C and 4C in mice (Fig. 3a, Supplementary Note 1 for all pairwise comparisons), and between 8C and ICM in humans (Fig. 3b, Supplementary Note 2 for all pairwise comparisons). In mice, we observed a shift of Obox-factor activity in 2C to an activation of Tead (Tead1-4) and AP-2 (Tfap2a/c/e) motifs in 4C. Notably, AP-2/Tfap2c is required for normal embryogenesis in mice[26] and was also recently shown to act as a chromatin modifier that opens enhancers proximal to pluripotency factors in human[27]. We observed a similar shift of TF activity for homeobox factors such as PITX1-3, RHOXF1, CRX, and DMBX1 at the human 8C stage towards higher scores in ICM for known pluripotency factors, such as POU5F1 (OCT4) and other POU-factors.

Throughout the pairwise comparisons, we observed that TFs from the same families often display similar binding kinetics within species, which is not surprising since they often possess highly similar binding motifs (Fig. 3a; right). To characterize TF similarity, TOBIAS clusters TFs based on the overlap of TFBS within investigated samples (Fig. 3c, d). This enables quantification of the similarity and clustering of individual TFs that appear to be active at the same time. Thereby, we observed a group of homeobox motifs, which cluster together with more than 50% overlap of their respective binding sites in mouse (Fig. 3c). In contrast, other TFs such as Tead and AP-2 cluster separately, indicating that these factors utilize independent motifs (the full tree is found in Supplementary Note 1). While this might appear trivial, this clustering of TFs in fact also highlights differences in motif usage between human and mouse. One prominent example is the RHOXF1 motif, which shows high binding-site overlap with Obox 1/3/5 and Otx2 binding sites in mouse (Fig. 3c; ~60% overlap), but does not cluster with OTX2 in human (Fig. 3d; ~35% overlap). This observation could suggest important functional differences of RHOX/Rhox TFs between mice and humans. In support of this hypothesis, *RHOXF1*, *RHOXF2*, and *RHOXF2B* are exclusively expressed at 8C and ICM in humans, whereas Rhox factors are not expressed in corresponding developmental stages of preimplantation in mice (expression values are given in the Source Data file). Conceivably, this observation, together with the finding that murine Obox factors share the same motif as RHOX-factors in humans, suggests that Obox TFs might function similarly to RHOX-factors during ZGA. Altogether, the TOBIAS-mediated TF clustering based on TFBS overlap allows for quantification of target-similarity and divergence of TF function between motif families.

**Dux expression induces massive changes in TF networks.** Throughout the investigations of human and mouse development we became particularly interested in the Dux/DUX4 TF, which TOBIAS predicted to be one of the earliest factors to be active in both organisms (Fig. 2b and Supplementary Fig. 4c). Interestingly, despite the fact that Dux has already been proved to play a prominent role in ZGA[2], there is still a poor understanding of how Dux regulates its primary downstream targets, and consequently its secondary targets, during this process. We therefore applied TOBIAS to identify Dux binding sites utilizing an ATAC-seq dataset of *Dux* overexpression (*Dux*OE) in mESC[2].

As expected, the differential TF activity predicted by TOBIAS showed an increase in activity of Dux and Obox TFs, as well as Hltf, which was already highlighted to be common between mouse and human ZGA (Fig. 4a). Interestingly, this was accompanied by a massive loss of TF binding for pluripotency markers, such as Nanog, Pou5f1 (OCT4), and Sox2 upon *Dux*OE, indicating that Dux renders previously accessible chromatin sites associated with pluripotency inaccessible.

Consistently, Dux footprints (Fig. 4b; left) were clearly evident upon *Dux*OE. In comparison to existing bias correction methods, we found TOBIAS to be better at uncovering this footprint between Control and *Dux*OE conditions (Supplementary Fig. 5a). Importantly, TOBIAS additionally discriminated ~30% of all potential binding sites within open chromatin regions to be bound in the *Dux*OE condition (Fig. 4b; right). To rank the biological relevance of the individually changed binding sites between control and *Dux*OE conditions, we linked all annotated gene loci to RNA expression. A striking correlation between the gain-of-footprint and gain-of-expression of corresponding loci was clearly observed and mirrored by the TOBIAS predicted bound/unbound state (Fig. 4c). Among the genes within the list of bound Dux binding sites were well-known Dux targets including

*Zscan4c* and *Pramef25*[28], for which local footprints for Dux were clearly visible (Fig. 4d). The high resolution of footprints is particularly pronounced for *Tdpoz1*, which harbors two potential Dux binding sites of which one is clearly footprinted in the score track, while the other is predicted to be unoccupied (Fig. 4d; bottom). In line with this, *Tdpoz1* expression is significantly upregulated upon *Dux*OE as revealed by RNA-seq (log2FC: 6,95). Consistently, *Tdpoz1* expression levels are highest at 2C and decrease thereafter, indicating that *Tdpoz1* is likely a direct target of Dux during PD both in vitro and in vivo. Footprinting scores also directly correlated with ChIP-seq peaks for Dux in the *Tdpoz1* promoter (Supplementary Fig. 5b), an observation which we also found at other positions (Supplementary Fig. 5c, d).

Many of the TOBIAS-predicted Dux targets encode TFs themselves. Therefore, we applied the TOBIAS network module to subset and match all activated binding sites to TF target genes with the aim of inferring how these TF activities might connect. Thereby, we could model an intriguing pseudo timed TF-activation network. This directed network predicted a TF-activation cascade initiated by Dux, resulting in the activation of 7 primary TFs which appear to subsequently activate 32 further TFs (first three layers depicted in Fig. 4e). As Dux is a regulator of ZGA, we asked how the in vitro activated Dux network compared to gene expression throughout PD in vivo. Strikingly, the in vivo RNA-seq data of the developmental stages[16] confirmed an early 2C specific expression of *Dux*, followed by a slightly shifted activation pattern for all direct Dux targets except for *Rxrg* (Fig. 4f). However, it is of note that *Rxrg* is significantly upregulated in the in vitro *Dux*OE from which the network is inferred (see Source Data for Fig. 4c), pointing to both the similarities and differences between the in vivo 2C and in vitro 2C-like stages induced by Dux. In conclusion, these data suggest that beyond identifying specific target genes of individual TFs, TOBIAS can promote biological insight by predicting entire TF-activation networks.

**Dux targets repeat elements.** Notably, many of the predicted Dux binding sites (40%) are not annotated to genes (Fig. 4g), raising the question what role these sites play in ZGA. Dux is known to induce expression of repeat regions such as long terminal repeats (LTRs)[2] and consistently, we found that more than half of the DUX-bound sites without annotation to genes are indeed located within known LTR sequences (Fig. 5a), which were transcribed both in vitro and in vivo (Fig. 5b; LTR). Interestingly, we additionally found that 28% of all non-annotated Dux binding sites overlap with genomic loci encoding LINE1 elements. Although LINE1 expression does not appear to be altered in mESC, there is a striking pattern of increasing LINE1 transcription from 4C–8C (Fig. 5b; LINE1) in vivo, pointing to a possible role of LINE1 regulation throughout PD. Finally, we found a portion of the Dux binding sites, which do not overlap with any annotated gene nor with putative regulatory repeat sequences, even though transcription clearly occurs at these sites (Fig. 5b; no overlap). One example is a predicted Dux binding site on chromosome 13, which coincides with a spliced region of increased expression between control mESC/*Dux*OE and comparable high expression in 2C, 4C, and 8C (Supplementary Fig. 6). These data suggest the existence of novel transcribed genetic elements, the function of which remains unknown, but which are likely controlled by Dux and may play a role during PD.

In conclusion, TOBIAS predicted the locations of Dux binding in promoters of target genes, and could propose how Dux initiates TF-activation networks and induces expression of repeat regions. Importantly, these data further show that TOBIAS

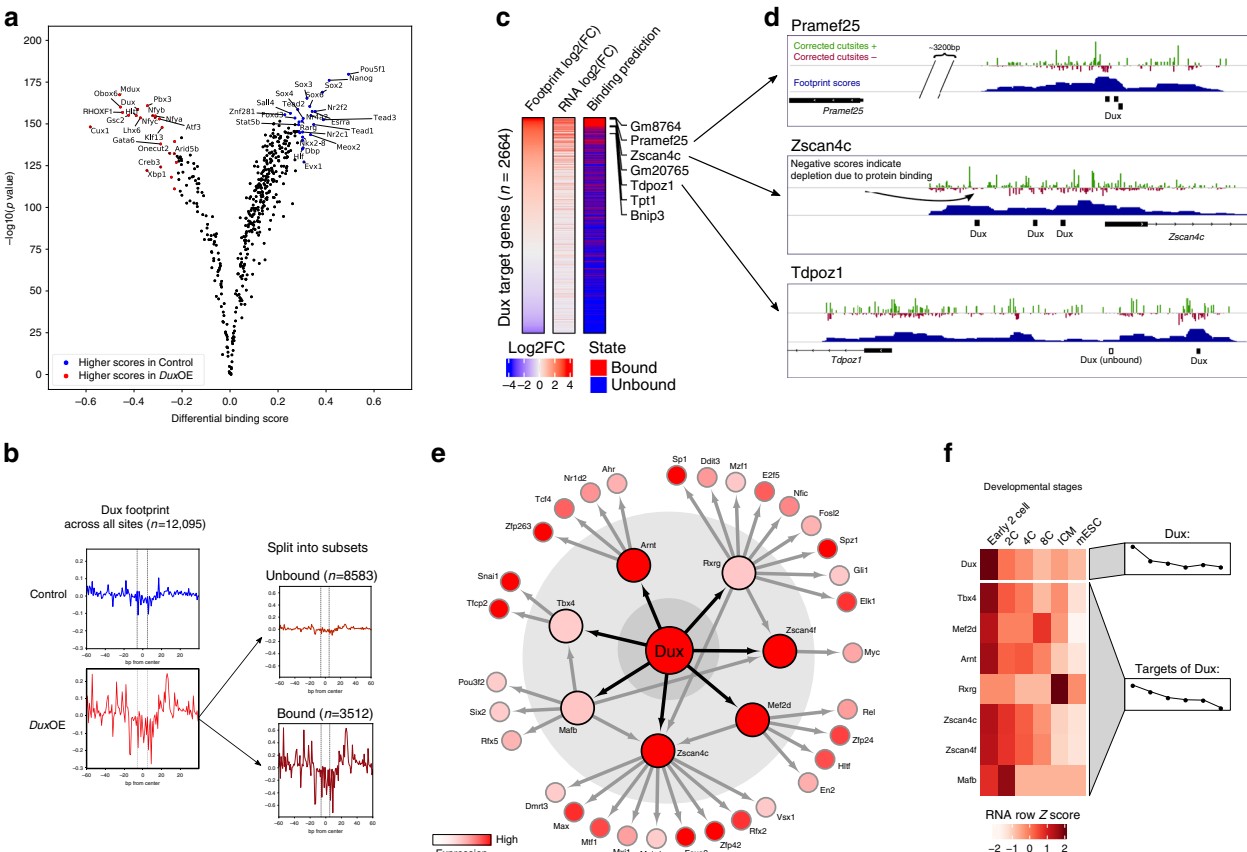

**Fig. 4 Dux binding induces transcription at gene promoters. a** Comparison of TF activities between mDux GFP- (Control; labeled in blue) and mDux GFP + (*Dux*OE; labeled in red). Volcano plot showing the TOBIAS differential binding score on the x-axis and -log10 (p value) on the y-axis; each dot represents one TF. **b** Aggregated footprint plots for Dux. The aggregated plots are centered on the predicted binding sites for Dux between Control and *Dux*OE conditions (left: all genomic sites). The total possible binding sites for *Dux*OE (*n* = 12,095) are separated into bound and unbound sites (right). The dashed lines represent the edges of the Dux motif. **c** Change in expression of genes near Dux binding sites. The heatmap shows *n* = 2664 Dux binding sites found in gene promoters. Footprint log2(FC) and RNA log2(FC) represent the matched changes between Control and *Dux*OE for footprints and gene expression, respectively. Log2(FC) is calculated as log2(*Dux*OE/Control). The column Binding prediction depicts whether the binding site was predicted by TOBIAS to be bound/unbound in the *Dux*OE condition. **d** Genomic tracks indicating three exemplary Dux binding sites and their target gene promoters and respective tracks for corrected cutsite signals (red/blue), TOBIAS footprint scores (blue), detected motifs (black boxes), and gene locations (solid black boxes with arrows indicating gene strand). **e** Dux transcription factor network. The TF-TF network is built of all TFBS with binding in TF promoters with increasing strength in *Dux*OE (log2(FC) > 0). Sizes of nodes represent the level of the network starting with Dux (Large: Dux, Medium: 1st level, Small: 2nd level). Nodes are colored based on corresponding RNA level in the *Dux*OE condition. Directed edges indicate binding sites in the respective gene promoter found by the TOBIAS CreateNetwork module. **f** Correlation of the Dux transcription factor network to expression during development. The heatmap depicts the in vivo gene expression during developmental stages. The right-hand group annotation highlights the difference in mean expression for each timepoint. The heatmap is split into Dux and target genes of Dux. Source data are available in the Source Data file.

predicts any TFBS with increased binding, not only those limited to annotated genes, which aids in uncovering novel regulatory genetic elements.

## Discussion

To the best of our knowledge, this is the first application of a DGF approach to visualize gain and loss of individual TF footprints in the context of time series, TF overexpression, and TF-DNA binding for a wide-range of TFs in parallel. Importantly, we found that these advances could in large part be attributed to the framework approach we took in developing TOBIAS, which enabled us to simultaneously compare global TF binding across samples and quantify changes in TF binding at specific loci. The modularity of the framework also allowed us to apply a multitude of downstream analysis tools to easily visualize footprints and gain even more information about TF binding dynamics as exemplified by the prediction of the Dux TF-activation network.

The power of this framework to handle time-series data becomes especially apparent when integrating the TOBIAS-based prediction of TF binding with RNA-seq data from the same timepoints. For instance, TOBIAS predicted that the maternally transferred TF SALL4 is active in 2C, while its gene expression pattern alone suggests later activation. While SALL4 was one of the TFs with the largest decrease in binding from 2C to 8C, it is, however, also worth noting that since TFs have different baseline activities, large changes between timepoints can also arise from very low activity scores. Although the scores are normalized towards global TF activity, differences in the quality of footprinting (due to sample-specific biases) can also influence the prediction of differential TF binding between conditions, and this should be considered as a limitation of this method. In this context, it is tempting to speculate that TFs for which footprinting scores are low, even though their RNA expression is high, might act as transcriptional repressors, because footprinting relies on the premise that TFs will increase chromatin

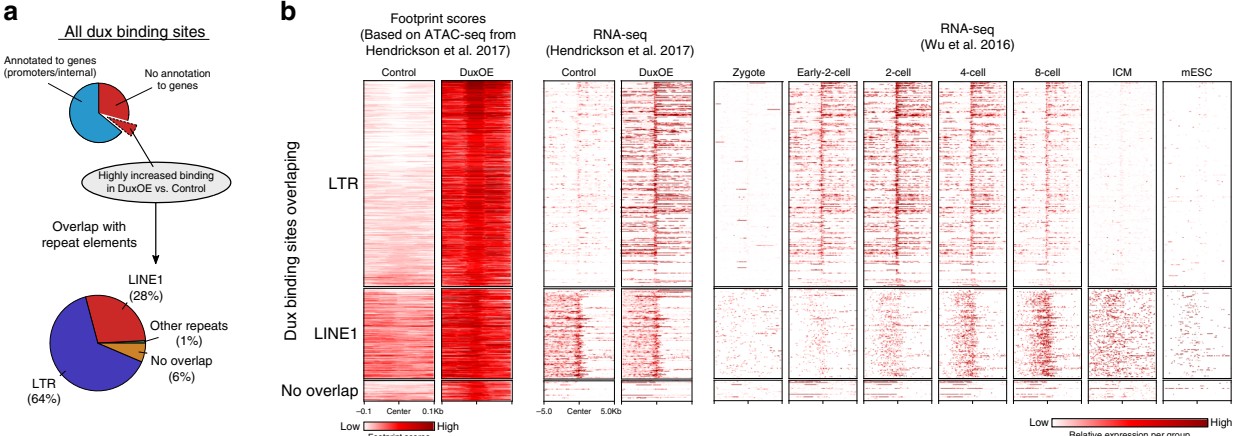

**Fig. 5 Dux binding influences expression of repeat elements. a** Dux binding sites overlap with repeat elements. All potential Dux binding sites are split into sites either overlapping promoters/genes or without annotation to any known genes (upper circle, blue/red). The bottom pie chart shows a subset of the latter, additionally having highly increased binding (log2(FC) > 1), annotated to repeat elements including LTR/LINE1 elements. **b** Dux induces expression of transcripts specific for preimplantation. Genomic signals for the Dux binding sites which are bound in *Dux*OE with log2(FC) footprint score >1 (i.e., upregulated in *Dux*OE) are split into overlapping either LTR, LINE1 or no known genetic elements (top to bottom); each row indicates one binding-site/associated gene loci. Footprint scores (±100 bp from Dux binding sites, left column) indicate the differential Dux binding between control and *Dux*OE (in vitro). RNA-seq shows the normalized read-counts from matched RNA-seq samples (center columns, in vitro) and throughout development (right columns, in vivo) within ±5 kb of the respective Dux binding sites. Dark red color indicates high expression.

accessibility around the binding site. In support of this hypothesis, recent investigations have suggested that repressors display a decreased footprinting effect in comparison to activators[29]. Therefore, the integration of ATAC-seq footprinting and RNA-seq is an important step in revealing additional information such as classification of TFs into repressors and activators, as well as the kinetics between expression and binding.

In the context of TF target prediction, we showed that TOBIAS could identify almost all known Dux targets. In addition to coding genes, our analysis disclosed novel Dux binding sites and significant footprint scores at LINE1 encoding genomic loci, which appear to be activated at the 4C/8C stage. This finding is especially interesting because a recent study has shown that LINE1 RNA can interact with Nucleolin and Kap1 to repress *Dux* expression[30]. Therefore, our findings suggest a kinetics driven model in which Dux not only initiates ZGA but also regulates its own termination by a temporally delayed negative feedback loop. How exactly this feedback loop could be controlled remains to be determined.

Despite the striking capability of DGF analysis, some limitations and dependencies of this method still remain. Among these is the need of high-quality TF motifs for matching footprint scores to individual TFs with high confidence. In other words, while the binding of a TF might create an effect that can be interpreted as a footprint, without a known motif, this effect cannot be matched to the corresponding TF. It also needs to be noted that footprinting analysis cannot take effects into account that arise from heterogeneous mixtures of cells wherein TFs are bound in some cells and in others not. Therefore, if not separated, the classification of differential binding will be an observation averaged across many cells, possibly masking subpopulation effects. Recent advances have enabled the application of ATAC-seq in single cells, but this generates sparse matrices, rendering footprinting approaches on single cells elusive. However, we speculate that by creating aggregated pseudo-bulk signals from large clustered single-cell ATAC datasets, DGF analysis might also become possible in single cells.

In conclusion, we present TOBIAS as the first comprehensive software that performs all steps of DGF analysis, natively supports multiple experimental conditions and performs visualization within one single framework. Although we utilized the process of PD as a proof of principle, the modularity and universal nature of the TOBIAS framework enables investigations of various biological conditions beyond PD. We believe that continued work in the field of DGF, including advances in both software and wet-lab methods, will validate this method as a resourceful tool to extend our understanding of a variety of epigenetic processes involving TF binding.

## Methods

**Processing of ATAC-seq data**. Raw sequencing fastq files were assessed for quality, adapter content and duplication rates with FastQC v0.11.7, trimmed using cutadapt[31] and aligned with STAR v2.6.0c[32] (parameters: --alignEndsType End-ToEnd --outFilterMismatchNoverLmax 0.1 --outFilterScoreMinOverLread 0.66 --outFilterMatchNminOverLread 0.66 --outFilterMatchNmin 20 --alignIntronMax 1 --alignSJDBoverhangMin 999 --alignEndsProtrude 10 --align-MatesGapMax 2000 --outMultimapperOrder Random --outFilterMultimapNmax 999 --outSAMmultNmax 1) to either the mouse or human genome using Mus_-musculus.GRCm38 or Homo_sapiens.GRCh38 versions from Ensembl[33]. Accessible regions were identified by peak calling for each sample separately using MACS2 (parameters: --nomodel --shift -100 --extsize 200 --broad)[34]. Peaks from each sample were merged to a set of union peaks across all conditions using bedtools merge. Each union peak was annotated to the transcriptional start site of genes (GENCODE[35]) in a distance of −10000/+1000 from the gene start using UROPA[36].

**Processing of RNA-seq data**. Raw reads were assessed for quality, adapter content and duplication rates with FastQC v0.11.7, trimmed using cutadapt[31] and aligned with STAR v2.6.0c[32] (parameters: --out-FilterMismatchNoverLmax 0.1 --outFilterScoreMinOverLread 0.9 --out-FilterMatchNminOverLread 0.9 --outFilterMatchNmin 20 --alignIntronMax 200000 --alignMatesGapMax 2000 --alignEndsProtrude 10 ConcordantPair --outMultimapperOrder Random --outFilterMultimapNmax 999) to either the mouse or human genome using Mus_musculus.GRCm38 or Homo_sapiens. GRCh38 versions from Ensembl[33].

**Processing of ChIP-seq data**. Raw sequencing files in fastq format were quality assessed by Trimmomatic by trimming reads after a quality drop below a mean of Q15 in a window of five nucleotides[37]. All reads longer than 15 nucleotides were aligned versus the mouse genome version mm10, keeping just unique alignments (parameters: --outFilterMismatchNoverLmax 0.2 --out-FilterScoreMinOverLread 0.66 --outFilterMatchNminOverLread 0.66 --out-FilterMatchNmin 20 --alignIntronMax 1 --alignSJDBoverhangMin 999

--outFilterMultimapNmax 1 --alignEndsProtrude 10 ConcordantPair) by using the STAR mapper[32]. Read deduplication was done by Picard (http://broadinstitute.github.io/picard/).

**Processing of TF motifs.** TF motifs were downloaded from JASPAR CORE 2018[38], the JASPAR PBM HOMEO collection and Hocomoco V11[39] databases. We further included the human ARGFX_3 motif from footprintDB[40] which originates from a HT-SELEX assay[41]. In addition to the Dux/Dux4 motifs of JASPAR and Hocomoco, we also included two TF motifs for Dux/DUX4 created using MEME-ChIP[42] with standard parameters on the ChIP-seq peaks of[28] (GSE87279).

JASPAR motifs were linked to Ensembl gene ids by mapping the provided Uniprot id to the Ensembl gene id through biomaRt[43]. Hocomoco motifs were likewise linked to genes through the provided HGNC/MGI annotation. Due to the redundancy of motifs between JASPAR and Hocomoco, we further filtered the TF motifs to one motif per gene, preferentially choosing motifs originating from mouse/human, respectively. For each TOBIAS run, we created sets of expressed TFs as estimated from RNA-seq in the respective conditions. This amounted to 590 motifs for the dataset on human preimplantation stages, 464 motifs for the dataset on mouse preimplantation, and 459 for the *Dux*OE dataset.

**Maternal genes.** Maternal genes for human and mouse were downloaded from the REGULATOR database[17]. Entrez gene ids were converted to Ensembl gene ids using biomaRt[43] and subsequently matched to available TF motifs as previously explained.

**Overlap of Dux binding sites with repeat elements.** Repeat elements for mm10 were downloaded from UCSC (http://hgdownload.cse.ucsc.edu/goldenpath/mm10/database/rmsk.txt.gz). Overlap of Dux sites to individual repeat elements was performed using bedtools intersect. The sum of overlaps were counted per repeat class (LINE1/LTR).

**Visualization.** All TF-score heatmaps were generated by R Version 3.5.3 and ComplexHeatmap package version 3.6[44]. Individual gene views were generated by loading TOBIAS output tracks into IGV version 2.6.2[45] or using the TOBIAS PlotTracks module, which is a wrapper for the svist4get visualization tool[46]. TF networks were drawn with Cytoscape version 3.7.1[47]. Heatmaps of genomic signal density were generated using Deeptools version 3.3.0[48]. All other figures, such as footprint plots, volcano plots and motif clustering dendrograms were generated by the TOBIAS visualization modules.

**The TOBIAS framework.** Details on the TOBIAS algorithms and framework setup are found in the Supplementary Methods part 1 and 2.

**Comparison of TOBIAS to existing methods.** Details on the validation and comparison of TOBIAS to existing methods for bias correction and footprinting are found in the Supplementary Methods part 3.

**Reporting summary.** Further information on research design is available in the Nature Research Reporting Summary linked to this article.

## Data availability

The source data for Figs. 2b, 4c, f, Supplementary Figs. 2c, e, f, g, 3, 4b, c, as well as expression values for *Rhox* and *Obox* genes throughout human and mouse development are available in the Source Data file. Raw ATAC-seq and RNA-seq data for human and mouse embryonic development are available from GEO under the accessions GSE66390 (mouse) and GSE101571 (human). Raw ATAC-seq, RNA-seq, and ChIP-seq data from *Dux* overexpression experiments are available from GEO under the accession GSE85632. Data for validation are available from ENCODE as explained in Supplementary Methods. Excerpts of the TOBIAS analysis results are accessible for dynamic visualization at: http://loosolab.mpi-bn.mpg.de/tobias-meets-wilson. UCSC track hubs (for viewing in the UCSC genome browser) of corrected Tn5 and footprint signals are available at: https://genome.ucsc.edu/cgi-bin/hgTracks?hubUrl=https://s3.mpi-bn.mpg.de/data-tobias-ucsc/hub.txt&genome=mm10 and https://genome.ucsc.edu/cgi-bin/hgTracks?hubUrl=https://s3.mpi-bn.mpg.de/data-tobias-ucsc/hub.txt&genome=hg38 for mouse and human respectively. All data are available from the authors upon reasonable request.

## Code availability

The TOBIAS software is publicly available at GitHub (https://github.com/loosolab/TOBIAS) and can additionally be obtained through PyPI and Bioconda.

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

## Acknowledgements

We would like to thank the IT-group at MPI-BN for continued support with IT-infrastructure. We would also like to thank Marius Dieckmann, the administrator of the Kubernetes cluster in the deNBi project at JLU (https://cloud.denbi.de/giessen/), for his support and help in implementing the TOBIAS-Nextflow Cloud version. This work was funded by the Max Planck Society, the German Research Foundation (DFG), grant KFO309 (project number 284237345, epigenetics core unit) to M.L, DZHK Rhine-Main Site, and by the Cardio-Pulmonary Institute (CPI), EXC 2026, Project ID: 390649896 to M.L. Open access funding provided by Projekt DEAL.

## Author contributions

M.B., C.K., J.K., and M.L. wrote the manuscript. M.B. developed the TOBIAS software. M.B., P.G., H.S., A.P., K.K., R.W., A.F., and J.P. performed additional bioinformatics analysis. T.B., J.K., and M.L. directed, coordinated, and supervised the work.

## Competing interests

The authors declare no competing interests.
