## [Peer Review File · Nature Communications]

REVIEWER COMMENTS

Reviewer #1 (Remarks to the Author):

In this study, Bentsen and colleagues introduce a new strategy (called TOBIAS) aimed at identifying footprints of chromatin-associated proteins from ATAC-seq accessibility profiles. They propose a data-driven correction of Tn5 bias and show that this is indeed increasing the signal-to-noise at individual footprints, thereby improving the inference of transcription factor (TF) binding from combining footprints and sequence predictions. TOBIAS also provides unique features such as comparing TF-activities in chromatin across multiple conditions (not just pairwise) and reconstructing TF-TF networks based on the predicted genome-wide binding profiles of single TFs. Besides being considerably faster than other available tools, I find it crucial that TOBIAS has also been well packaged and documented (<https://github.com/loosolab/TOBIAS>).

The authors first show that TOBIAS performs better than state-of-the-art tools, with the exception of msCentipede, to which TOBIAS performs comparably but at a fraction of the computational cost and the running time. The authors then used published data from both human and mouse ZGA to showcase the potential of TOBIAS to not only confirm previous observations but also to identify novel correlations in the data (such as those in Fig. 4e, 4h and 2c).

I think TOBIAS will prove an extremely valuable and accessible tool to characterize the activities of transcription factors across different conditions and will very much benefit researchers in many fields that involve the study of transcriptional regulation and chromatin processes. I think the manuscript should be considered for publication in Nature Communications, as long as the authors are able to address some specific concerns (see bullet-points below).

Major comments:

- Despite not being an expert in the field, I appreciate the resource value of the manuscript to the community studying ZGA, so I was surprised the authors did not provide any easy way to access the predicted footprints genome-wide (for example as track hubs for the UCSC genome browser - one for human and one for mouse). Have I missed this? If not, I think the authors should consider setting this up.

- Line 133: "TOBIAS generated a measurable footprint for 64% of the TFs (Suppl. Figure 3e). This is in contrast to previous reports wherein it has been suggested that only 20% of all TFs leave measurable footprints". This could be a very interesting observation, although fitting a two-component model to the distribution shown in Suppl. Fig. 3e does not seem a strong estimate to distinguish "measurable" footprints from the rest. Can the authors provide a more robust estimate, separately for each TF, for the corrected footprint depth? (e.g. generating TF-specific null distributions of average footprints to then calculate the FDR of each estimate shown in the distribution).

- The dynamic ranges of differences reported in transcriptional activities of many of the TFs in the human dataset (Suppl. Fig. 4 and Suppl. Table 2) are very small. Despite most of the inferences being made from these patterns of activity recapitulate previous knowledge in the field, when looking at specific instances (such as SALL4; lines 191-200) it is hard to evaluate if these differences are statistically significant. In the case of SALL4 for example, the on-off-on pattern might well be a flat one, given the absence of any assessment of variability for the activity scores in the current analysis. Can the authors provide an estimate of confidence intervals for these (and maybe all) activity scores, or at least explicitly discuss this limitation? The authors are making a strong case for this SALL4 observation in the Discussion, but I find the evidence supporting it rather weak.

Minor comments:

- Considering Suppl. Fig. 2a (and line 108): can the authors also provide panels showing

quantifications of the footprint depths? Although these observations are formally supported by the results shown in Suppl. Fig. 2b-c, this would strengthen their initial statement about TOBIAS clearly outperforming the rest of tools.

- With reference to this sentence from lines 208-209: "including several homeobox factors which already have described functions within ZGA" - can the authors specify the exact number of TFs (instead of "several") and highlight the corresponding TF-genes in the figure panel?

- I find it confusing that the authors first state that "The timing of ZGA varies between mice (2C) and humans (4C to 8C)" but then in Fig. 3 they focus the analyses on the 2C->4C transition in mouse and on the 8C->ICM in human. It seems that these comparisons are run to show "the ability of TOBIAS to capture differentially bound TFs, not only across the whole timeline, but also between individual conditions and species" but this is only mentioned after the analyses are introduced and commented. If this is the case, can the authors mention this beforehand? I think this would increase the clarity of the paragraph.

- With reference to this paragraph "Footprinting (TOBIAS ScoreBigwig module)" (line 544 in the Methods section): considering the notation in the previous paragraph, would it be more appropriate to use c_i instead of x_i (corrected number of cuts instead of the total?)

Reviewer #2 (Remarks to the Author):

Over the last years it has become apparent that in the accessible regions from DNase-seq or ATAC-seq data one can see footprints of transcription factors. This technique is generally referred to as digital genomic footprinting (DGF). The present manuscript introduces a pipeline ("TOBIAS") for DGF and applies this pipeline to analyze ATAC-seq data describing zygotic preimplantation development, based on publicly available data.

The TOBIAS pipeline appears to produce qualitatively very good results. It is fast, and offers a number of useful features like differential binding and network construction. However, the manuscript suffers from numerous shortcomings.

Firstly, the manuscript puts more emphasis on self-praise than on clarity of exposition. Even the first paragraph of the Results Section reads more like advertisement and does not convey much scientific information. The actual method behind TOBIAS is explained only inadequately. This reviewer spent quite some time studying Fig 1 which is where the method is supposed to be described (why is the overview of early embryonic development part of this figure?). Unfortunately, there is virtually no text to explain the extremely dense figure. The heart of the method seems to be what is depicted in part c, under "Bias correction and footprinting". What is the meaning of the insert to the left ("footprinting"), what is "Depth", what is the bottom image (where it says "threshold")?

The authors stress that TOBIAS corrects for sequence bias introduced to ATAC-seq by the transposase. However, other programs also claim to correct for this, and therefore it would be interesting to structure the validation accordingly and dissect whether the present advance is due to better bias correction or due to some other novel feature of the footprinting. As it stands, this is impossible for the reader to discern. Fig. S2 claims it would clarify this, but there isn't even a discussion of Fig S2a. It pertains to two TFs. What about the others? Can one quantify this?

S2b contains a group of distributions of auROC. The legend only says "auROC based on ENCODE ...". Which exact quantities are compared in the ROC curves?

Other figures and figure legends suffer from a similar lack of explanation and discussion.

The other novel features of TOBIAS appear useful but are not validated. Their proof is in the application to the ZGA data.

The description of the methods as given under "The TOBIAS framework", lines 501 onward, is unclear. The formulae are cryptic due to the unconventional notation combining words (flank, mid, total, overlap, dist) and groups of letters (wf, wn, Wn?,...) . Needs clear notation, a sketch to illustrate the quantities, and/or a list of the different symbols.

The analysis of the ZGA data is interesting, but follows the scheme of "We rediscover many known things therefore the rest of our findings also has to be true". The figures illustrating and supporting this analysis are again extremely dense and only superficially explained.

The first section under Results should just be called "Validation of TOBIAS". Why "Classification and ..."? What does "classification" mean in this context?

In summary, while TOBIAS is probably a good and useful tool, the manuscript fails in explaining the science behind it.

Reviewer #1

1: Despite not being an expert in the field, I appreciate the resource value of the manuscript to the community studying ZGA, so I was surprised the authors did not provide any easy way to access the predicted footprints genome-wide (for example as track hubs for the UCSC genome browser - one for human and one for mouse). Have I missed this? If not, I think the authors should consider setting this up.

We thank the reviewer for this positive assessment of our use case data, and as suggested, we have set up a UCSC Hub containing Tn5 insertion tracks and footprint scores for the mouse and human preimplantation stages analyzed in our manuscript. The links to these are: <https://genome.ucsc.edu/cgi-bin/hgTracks?hubUrl=https://s3.mpi-bn.mpg.de/data-tobias-ucsc/hub.txt&genome=mm10>

for mouse and

<https://genome.ucsc.edu/cgi-bin/hgTracks?hubUrl=https://s3.mpi-bn.mpg.de/data-tobias-ucsc/hub.txt&genome=hg38>

for human data. These links have also been added to the “Data availability”-section. Additionally, we would like to highlight the availability of the differential footprint analysis for both human and mouse, which are available to the community at <http://loosolab.mpi-bn.mpg.de/tobias-meets-wilson>.

2: Line 133: “TOBIAS generated a measurable footprint for 64% of the TFs (Supp. Figure 3e). This is in contrast to previous reports wherein it has been suggested that only 20% of all TFs leave measurable footprints”. This could be a very interesting observation, although fitting a two-component model to the distribution shown in Suppl. Fig. 3e does not seem a strong estimate to distinguish “measurable” footprints from the rest. Can the authors provide a more robust estimate, separately for each TF, for the corrected footprint depth? (e.g. generating TF-specific null distributions of average footprints to then calculate the FDR of each estimate shown in the distribution).

We are convinced that the increased footprint visibility highlights bias correction as a very important feature of TOBIAS, especially when comparing our results with the investigations of previous literature. The original statement of “only 20% of transcription factors create measurable footprints”, which was put forward by Baek et al. 2017, was estimated using a two-component mixture model of footprint depths. To make our results comparable with this previous statement, we therefore chose to apply the same measure of footprint depth as Baek et al. did. However, we agree with the reviewer regarding the arguable robustness of this model.

In order to address the reviewers’ comment, we calculated a secondary estimate of “measurable” footprints as now presented in Supp. Figure 2g. As suggested by the reviewer, we have calculated a null-distribution of aggregate footprints per transcription factor, and used these to calculate the significance of the observed footprint depths. The results of this

analysis confirms that across the four cell types used for validation, the rate of measurable footprints is in the range of 65-70%. We have further updated the two-component mixture model (Supp. Figure 2f; previously Supp. Figure 3e) with the GM12878 cell line, in order to make this estimate comparable with the new Supp Figure 2g. We thank the reviewer for this comment, as we feel that the additional work has strengthened this point of the validation.

3: The dynamic ranges of differences reported in transcriptional activities of many of the TFs in the human dataset (Suppl. Fig. 4 and Suppl. Table 2) are very small. Despite most of the inferences being made from these patterns of activity recapitulate previous knowledge in the field, when looking at specific instances (such as SALL4; lines 191-200) it is hard to evaluate if these differences are statistically significant. In the case of SALL4 for example, the on-off-on pattern might well be a flat one, given the absence of any assessment of variability for the activity scores in the current analysis. Can the authors provide an estimate of confidence intervals for these (and maybe all) activity scores, or at least explicitly discuss this limitation? The authors are making a strong case for this SALL4 observation in the Discussion, but I find the evidence supporting it rather weak.

We thank the reviewer for their comment and agree with the lack of assessment of the variability of activity scores. Due to the differences in ‘footprintability’ and activity of different TFs, and because we only have 4 time points, it is difficult to provide confidence intervals for these scores. Instead, we have calculated in what quantile the SALL4 changes reside in comparison to the other TFs, and found that it is within the 7% largest decreasing TFs between 2C and 8C. However, as the reviewer has suggested, we have also mentioned this limitation in the discussion, and have rephrased the results section.

4: Considering Suppl. Fig. 2a (and line 108): can the authors also provide panels showing quantifications of the footprint depths? Although these observations are formally supported by the results shown in Suppl. Fig. 2b-c, this would strengthen their initial statement about TOBIAS clearly outperforming the rest of tools.

According to the reviewers request, we have added additional quantifications of the footprint depths in Supp. Figure 2c. This analysis quantifies what is shown in Supp. Figure 2a, namely that the ‘TOBIAS ATACorrect’ method has significantly better separation of bound and unbound subsets of TFBS when quantified using the FPD metric.

5: With reference to this sentence from lines 208-209: “including several homeobox factors which already have described functions within ZGA” - can the authors specify the exact number of TFs (instead of “several”) and highlight the corresponding TF-genes in the figure panel?

We have changed Figure 2c according to the reviewer's advice and labeled known ZGA related TFs. In addition, we rephrased the respective sentence to "(...) including OTX1, GSC, CPHX and HLTF, which already have described functions within ZGA".

6: I find it confusing that the authors first state that "The timing of ZGA varies between mice (2C) and humans (4C to 8C)" but then in Fig. 3 they focus the analyses on the 2C->4C transition in mouse and on the 8C->ICM in human. It seems that these comparisons are run to show "the ability of TOBIAS to capture differentially bound TFs, not only across the whole timeline, but also between individual conditions and species" but this is only mentioned after the analyses are introduced and commented. If this is the case, can the authors mention this beforehand? I think this would increase the clarity of the paragraph.

The comparisons are intended to showcase the information gained from the pairwise comparison module and respective plots as created by TOBIAS. We have tried to present these aspects by first looking at the data available during ZGA in mouse and human individually (in line with the timepoints available), followed by an inter-species comparison in order to illustrate common signatures given at the onset of ZGA. Because the start of ZGA is less defined in humans, we have chosen to look at the 8C time point as the start of ZGA. Therefore, both 2C-4C (mouse) and 8C-ICM (human) represent the transition between the event of ZGA and the subsequent development. According to the reviewer's advice, we have made changes to this paragraph to make the motivation behind this selection more clear.

7: With reference to this paragraph "Footprinting (TOBIAS ScoreBigwig module)" (line 544 in the Methods section): considering the notation in the previous paragraph, would it be more appropriate to use c_i instead of x_i (corrected number of cuts instead of the total?)

We apologize for the inconsistency in notation in the methods section. We have improved this and other mathematical notations in the methods.

Reviewer #2

1: Firstly, the manuscript puts more emphasis on self-praise than on clarity of exposition. Even the first paragraph of the Results Section reads more like advertisement and does not convey much scientific information. The actual method behind TOBIAS is explained only inadequately.

We thank the reviewer for their feedback and agree that the first part of the results can come across as too focused on the features of TOBIAS. We have toned down the self-praise as recommended by the reviewer and have added more explanations on the individual TOBIAS workflow steps, especially those given in Figure 1 (see also answer to comment 2). We have also moved Table 1 to Supplementary Table 1. Regarding the lack of method explanation raised by the reviewer, we want to point out that the TOBIAS framework consists of multiple steps, which contain considerations of both mathematical models and software related aspects (e.g. pipelining, cloud computing etc.). As these details might not be of interest to all readers, and would considerably reduce the flow of illustrating the potential of DGF, we have chosen to present the main concepts of TOBIAS in the main text (as presented in Figure 1). In order to address the interests of a reader from the field of computational biology, we refer to the detailed information on the algorithms and software execution aspects available in the methods section, as well as on the TOBIAS Github repository page. In order to improve clarity of the TOBIAS methods and to comply with the reviewer's critiques, we have restructured and rephrased the methods section according to the reviewer's advice (see also answer to comment 6).

2: This reviewer spent quite some time studying Fig 1 which is where the method is supposed to be described (why is the overview of early embryonic development part of this figure?). Unfortunately, there is virtually no text to explain the extremely dense figure. The heart of the method seems to be what is depicted in part c, under "Bias correction and footprinting". What is the meaning of the insert to the left ("footprinting"), what is "Depth", what is the bottom image (where it says "threshold")?

We appreciate the concerns of the reviewer regarding the density of Figure 1. Because not all readers are familiar with the topic of early embryonic development, we introduce the basics quite early in the introduction of the paper, and feel that it is important to give a simplified overview of the developmental stages within Figure 1. However, we agree that Figure 1c is very dense. In order to decrease the density of the figure and to improve clarity, we have divided the previous Figure 1c into multiple subparts and have extended the explanations of the methodology in the manuscript, as well as in the figure legend. Additionally, we have restructured the section on validation of TOBIAS to emphasize the different parts of the algorithm, particularly the "bias correction" and "footprinting" modules (see also answer to comment 3). Further we reduced the complexity of the TOBIAS downstream section in

Figure 1. We are confident that these changes have improved the reader's understanding of the figure.

3: The authors stress that TOBIAS corrects for sequence bias introduced to ATAC-seq by the transposase. However, other programs also claim to correct for this, and therefore it would be interesting to structure the validation accordingly and dissect whether the present advance is due to better bias correction or due to some other novel feature of the footprinting. As it stands, this is impossible for the reader to discern. Fig. S2 claims it would clarify this, but there isn't even a discussion of Fig S2a. It pertains to two TFs. What about the others? Can one quantify this?

When validating TOBIAS, we were faced with the problem that not all footprinting tools provide bias correction (or even the possibility to run the algorithm on bias corrected data), and it was therefore difficult to dissect which features of footprinting provided the largest effect. In order to examine this question, and to test whether the advances of TOBIAS are due to bias correction alone, or due to other novel features of the footprinting, we have performed a comparison of the original footprint occupancy score (FOS) (depletion = footprint) for uncorrected and corrected Tn5 signals and compared these results to the TOBIAS score (depletion+accessibility = footprint). These new investigations are presented in Supp. Figure 3f-h and are discussed in the section "Validation of TOBIAS footprinting". While we find a small increase in the mean auROC of the FOS score on corrected signals vs. uncorrected signals, there is a significant increase in the auROC when comparing the FOS score to the TOBIAS footprinting score. This indicates that the protein-bound TFBS at specific genomic loci can be too sparse to show a canonical footprint shape, and that there is a clear benefit of the TOBIAS score which also weights accessibility within the score.

Regarding the visualization of aggregated footprints, we have extended the validations by additional analysis as shown in Supp Figure 2. As suggested by the reviewer, we have added a quantification of the footprint depths (Supp. Figure 2c). This analysis specifically quantifies what is shown in Supp. Figure 2a, which is that the 'TOBIAS ATACCorrect' has significantly better separation of bound and unbound subsets of TFBS when quantified using the footprint depth (FPD) metric. While Supp. Figure 2a only highlights two TFs, we provide a visualization of all individual TF footprints before and after correction with all bias correction tools in Supplementary File 1. While we agree with the reviewer that the lack of additional quantification made it hard for the reader to discern the results of this figure, we are confident that the additions to Supp. Figure 2 now addresses this issue.

4: S2b contains a group of distributions of auROC. The legend only says "auROC based on ENCODE ...". Which exact quantities are compared in the ROC curves? Other figures and figure legends suffer from a similar lack of explanation and discussion.

We carefully reviewed the legend of S2b as well as other figures and have improved the legends. We have also added more references to the methods where the calculation of auROC is explained in greater detail.

5: The other novel features of TOBIAS appear useful but are not validated. Their proof is in the application to the ZGA data.

When preparing this manuscript, we have had long discussions regarding the optimal way to present and validate the results of our tool. With a complex method such as DGF, it was difficult to identify a good use-case scenario which would allow the validation using standard methods such as ChIP-seq, while also showcasing the potential of DGF when applied to a complex dataset including multiple conditions, organisms and time points of interest. Therefore, we have chosen to perform an in-depth validation on data from four well-studied human cell lines with matching ChIP-seq assays (summarized by Supp. Figure 2 and 3) to fulfill the first task. That way, we were able to compare TOBIAS to existing tools for bias correction and footprinting, and we believe that the results of these investigations validate the strength and accuracy of TOBIAS analysis on ATAC-seq data. For the latter task, we utilized zygotic genome activation and early mammalian development as a show-case, and we agree with the reviewer that we have not been able to validate all of our results for this part. Nevertheless, we defined some basic assumptions about footprinting analysis, which we believe can still serve as indirect evidence supporting the results:

1. One of the basic assumptions about footprinting is that the expression and binding of a TF should correspond to the gain of a footprint for the corresponding TF. Although this aspect sounds trivial, proving it is not, and to the best of our knowledge, this basic assumption was not yet addressed in other applications of DGF. We believe that this is because of the multitude of parameters influencing the visibility of footprints, particularly including bias correction and biological background, which we have taken into consideration in this paper. In the method validation (Supp. Figure 2), in the footprints from the developmental timeline (Figure 2b), and in the investigation of Dux (Figure 4b), we have consistently shown the gain, and also subsequent loss, of footprints in vivo and in vitro. Particularly, we find that the application of TOBIAS to the Dux overexpression data is a remarkable proof-of-concept, as it illustrates the basic assumption that a TF cannot create a footprint if it is not expressed. As shown in Figure 4b, we found this assumption to be true, and notably, the footprint was not detectable by the two other available methods for ATAC-seq correction (see Supp. Figure 5a).
2. We have assumed that if footprinting analysis uncovers TF binding, there should also be other lines of evidence supporting these observations. Therefore, we incorporated results of other assays, such as RNA-seq and ChIP-seq if available. For instance, we have provided a correlation of

TOBIAS scores and RNA-seq as provided in Supp Figure 4b and Supp Table 3. In the context of Figure 2b, DUX4, ZSCAN4 and GATA2 all show a high correlation between the visibility of the footprint and the expression of the RNA itself. In Figure 4c, we show a high correlation between footprint scores and the expression of target genes as defined by RNA-seq. We also show examples of transcription close to some potential Dux bound target loci that were not yet described, but can be supported by public RNA-seq or ChIP-seq data (Figure 4d, Supp Figure 5b-d and Supp. Figure 6). Therefore, although we have not verified the presence/binding of all TFs at the given time points, we believe that the correlation to multiple other sources of data support the results.

3. Finally, we have made use of previous literature to select biologically meaningful TFs throughout our analysis. We have used prior knowledge of ZGA to check whether expected “behavior” fit to the TOBIAS analysis derived from ATAC-seq data. As an example, we selected ZSCAN4 as one of the very few accepted target genes of Dux and were able to show a gain and loss of a footprint exactly when it would be expected during the developmental timeline (Figure 2b). In the mention of the role of Obox TFs, and their overlap with RHOX binding sites, we have discussed the known biological functions of these TFs in both human and mouse. Additionally, the global TF activity landscape generated by TOBIAS (e.g. Figure 2a) matched the idea that there are distinct groups of TFs which drive the stages of ZGA.

Summarizing, we agree with the reviewer that we have used ZGA as an example to apply and prove our DGF framework. However, we are convinced to have used a biological and methodical convincing setup, taking into account the complete start/shutdown of TFs in vitro and in vivo, accompanied with very distinct activity patterns of individual TFs. We believe that these investigations serve as a demonstration (rather than a validation) of the TOBIAS functionalities.

6: The description of the methods as given under "The TOBIAS framework", lines 501 onward, is unclear. The formulae are cryptic due to the unconventional notation combining words (flank, mid, total, overlap, dist) and groups of letters (wf, wn, Wn?,...) . Needs clear notation, a sketch to illustrate the quantities, and/or a list of the different symbols.

We apologize for the inconsistency of the mathematical notations. We have carefully reviewed the methods and have provided better notation and explanations of the different symbols. For the calculation of the TOBIAS footprint score, we have additionally added a sketch of the quantities (Supp. Figure 3f). We are confident that these changes have clarified the methodology.

7: The analysis of the ZGA data is interesting, but follows the scheme of "We rediscover many known things therefore the rest of our findings also has to be true". The figures illustrating and supporting this analysis are again extremely dense and only superficially explained.

In line with our answer to the reviewers earlier question (comment 5), we want to point out that we do not claim to prove all findings described for the ZGA data (particularly Figure 3 and 4). We are aware that it would require further in-depth investigations including additional wet-lab experiments to fully verify the biological importance of our observations, which is far beyond the scope of this paper. ZGA is rather a showcase of the potential of DGF when applied to a complex biological background. One example which we have investigated but not yet validated is the introduced hypothesis of a Dux feedback loop driven by activation of LINE elements. Their role in the context of shutting down Dux expression was recently described by Percharde, 2018, but the regulation of LINE elements in the context of ZGA is still elusive. Further experiments in the laboratory will be necessary to prove and unravel the exact mechanisms, but are beyond the scope of this paper.

While we agree with the reviewer that not all findings are true just because some are, we believe that verifying a subset of findings by utilizing given knowledge can strengthen the plausibility of the rest. We carefully reviewed our manuscript and have made changes to improve the distinction between validation and correlation of findings.

8: The first section under Results should just be called "Validation of TOBIAS". Why "Classification and ..."? What does "classification" mean in this context?

To improve the presentation of the TOBIAS algorithm, and in line with comment 3, we have split this section into two parts. The first one now handles the Tn5 bias correction, while the second one is focused on the validation of the TOBIAS footprinting. The two headings for these sections are renamed to "Impact of TOBIAS bias correction on footprint visibility" and "Validation of TOBIAS footprinting" as the reviewer had advised.

9: In summary, while TOBIAS is probably a good and useful tool, the manuscript fails in explaining the science behind it.

We thank the reviewer for this positive feedback regarding our tool. We are confident that with the help of the reviewer's comments and advice, we were able to improve the explanation of the method and science behind it, as well as provide clarity on the exemplary application to ZGA.

REVIEWERS' COMMENTS:

Reviewer #1 (Remarks to the Author):

The authors addressed all the relevant concerns raised by the reviewers. Notably, the availability to the community of the results shown in the manuscript has also been improved. I think this newer version of the manuscript is suitable for publication in Nature Communications. I only have very few minor comments to improve the accuracy of some sentences and/or the overall clarity:

- Line 161: "Of note, the TOBIAS 'ATACorrect' module applies an "expected" intermediate." I understand the message of this sentence by reading the overall paragraph that follows, but per se I could not understand the meaning of it;
- Line 499: "mechanisms" rather than "concepts"?
- Considering the new versions of figs. 1 and 2, I now see panel 1A as a better fit for fig. 2 (so that fig. 1 would be completely focused on introducing TOBIAS, and fig. 2 on the ZGA analysis).

Reviewer #2 (Remarks to the Author):

All of the reviewer's concerns have been addressed. The paper has improved greatly. It is interesting and reads well. The new method and the results presented constitute a significant scientific advance.

The sentence in line 119 is unclear: "expected" intermediate ?

REVIEWERS' COMMENTS

Reviewer #1 (Remarks to the Author):

The authors addressed all the relevant concerns raised by the reviewers. Notably, the availability to the community of the results shown in the manuscript has also been improved. I think this newer version of the manuscript is suitable for publication in Nature Communications. I only have very few minor comments to improve the accuracy of some sentences and/or the overall clarity:

- Line 161: "Of note, the TOBIAS 'ATACorrect' module applies an "expected" intermediate." I understand the message of this sentence by reading the overall paragraph that follows, but per se I could not understand the meaning of it;

- Line 499: "mechanisms" rather than "concepts"?

- Considering the new versions of figs. 1 and 2, I now see panel 1A as a better fit for fig. 2 (so that fig. 1 would be completely focused on introducing TOBIAS, and fig. 2 on the ZGA analysis).

Author's answer:

We have made the explanation of the expected intermediate more clear in the text. Regarding the comment on line 499, this part has been joined with the discussion and this line therefore no longer exists. We followed to the reviewers advice regarding figure 1a and have moved this to figure 2 (now figure 2a). Finally, we would like to thank the reviewer for their constructive comments throughout the entire review process.

Reviewer #2 (Remarks to the Author):

All of the reviewer's concerns have been addressed. The paper has improved greatly. It is interesting and reads well. The new method and the results presented constitute a significant scientific advance.

The sentence in line 119 is unclear: "expected" intermediate ?

Author's answer:

We have made the explanation of the expected intermediate more clear in the text. We would like to thank the reviewer for their helpful comments throughout the entire review process.